

# Processing reflectivity and Doppler velocity from EarthCARE's cloud profiling radar: the C-FMR, C-CD and C-APC products

Pavlos Kollias[1,2], Bernat Puidgomènech Treserras[2], Alessandro Battaglia[3,4,5], Paloma C. Borque[2], and Aleksandra Tatarevic[6]

[1]Division of Atmospheric Sciences, Stony Brook University, NY, USA
[2]Department of Atmospheric and Oceanic Sciences, McGill University, Montreal, Canada
[3]Politecnico of Turin, Turin, Italy
[4]Department of Physics and Astronomy, University of Leicester, Leicester, UK
[5]National Centre for Earth Observation, Leicester, UK
[6]Meteorogical Research Division, Environment and Climate Change Canada, Dorval, QC, Canada

**Correspondence:** Pavlos Kollias (pavlos.kollias@stonybrook.edu)

**Abstract.** The Earth Clouds, Aerosols and Radiation (EarthCARE) satellite mission is a joint effort by the European Space Agency (ESA) and the Japanese Aerospace Exploration Agency (JAXA). The EarthCARE mission features the first spaceborne 94-GHz Cloud Profiling Radar (CPR) with Doppler capability. The raw CPR observations and auxiliary information are used as input to three L2 algorithms 1) C-APC: Antenna Pointing Characterization, 2) C-FMR: CPR feature mask and reflectivity and 3) C-CD: Corrected CPR Doppler Measurements. These algorithms apply quality control and corrections to the CPR primary measurements and derive important geophysical variables such as hydrometeor locations, and best estimates of particle sedimentation fall velocities. The C-APC algorithm uses natural targets to introduce any corrections needed to the CPR raw Doppler velocities due to the CPR antenna pointing. The C-FMR product provides the feature mask based on only-reflectivity CPR measurements and quality controlled radar reflectivity profiles corrected for gaseous attenuation at 94 GHz. In addition, C-FMR provides best estimates of the Path Integrated Attenuation (PIA) and flags identifying the presence of multiple scattering in the CPR observations. Finally, the C-CD product provides the quality-controlled, bias-corrected mean Doppler velocity estimates (Doppler measurements corrected for antenna mis-pointing, non-uniform beam filling, and velocity folding). In addition, the best estimate of the particle sedimentation velocity is estimated using a novel technique.

## 1 Introduction

Spaceborne active and passive instruments are key to obtain a holistic global picture of cloud and aerosol vertical properties. The National Aeronautics and Space Administration (NASA) A-Train constellation of satellites first demonstrate the synergy and effectiveness of using such kind of measurements. In particular, measurements from three satellites: CloudSat (with its 94-GHz Cloud Profiling Radar, Stephens et al. (2002)), CALIPSO (with its Cloud and Aerosols Lidar with Orthogonal Polarization, Winker et al. (2007) ) and Aqua (with both narrow-band and broad-band passive radiometers, Schoeberl et al. (2006)).

Following this heritage, the Earth Clouds, Aerosol and Radiation Explorer (EarthCARE) mission developed by the European Space Agency (ESA) and Japan Aerospace Exploration Agency (JAXA) is scheduled for launch in 2024 (Illingworth et al.,



2015). The EarthCARE mission was designed with the three instruments on the same platform in order to maximize the benefit that may be realized by combining the different sensors. One of the instruments onboard the EarthCARE satellite is a high sensitivity 94-GHz Cloud Profiling Radar (CPR) with Doppler capability (Kollias et al., 2014). The EarthCARE CPR is the

second 94-GHz radar in space after NASA's CloudSat. The EarthCARE CPR uses a larger antenna (2.5 m compare to 1.6 m diameter for CloudSat) and operates at lower altitude (400 km versus 710 km for CloudSat). As a result, the EarthCARE CPR (hereafter EC-CPR) exhibits higher sensitivity (-36 dBZ versus -29 dBZ for CloudSat) and it is the first atmospheric radar in space with Doppler velocity measurement capability (Kollias et al., 2018, 2022). A comprehensive list of L2a (single instrument) and L2B (synergistic) data products has been designed and implemented to achieve the EarthCARE mission scientific

objectivites. These products provide best estimates of aerosol, clouds and precipitation properties (Illingworth et al., 2015).

Here, the theoretical physical basis, the algorithm flow and structure of three L2a EC-CPR products (C-FMR, C-APC, and C-CD) is described. While there is a lot of heritage and experience in the development of the C-FMR from CloudSat (Mace et al., 2007; Haynes et al., 2009), the other two products (C-ATC and C-CD) address the quality control and interpretation of the first spaceborne, atmospheric Doppler radar measurements from space. Three high resolution model scenes generated by

the Environment and Climate Change Canada (ECCC) Global Environmental Multiscale (GEM) model are used to evaluate the performance of the EC-CPR data products in a wide range of cloud and precipitation conditions. The ECCC scenes and the forward simulated EarthCARE fields are available in (van Zadelhoff et al., 2022)

## 2 Background

### 2.1 CPR On board processing and the JAXA L1b C-NOM product

The JAXA CPR L1b product provides the input variables to the C-CD algorithm. The EC-CPR receiver has a logarithmic detector that is used to estimate the received echo power Pr (W) that is converted to radar reflectivity factor using radar calibration constant $C$ that is determined by the internal receiver calibration based on hot/cold input noise source. The procedure is very similar to that used in the CloudSat CPR (Tanelli et al., 2008). After the EarthCARE launch, the CPR calibration constant $C$ will be monitored using routine measurements of the ocean-surface return using the Li et al. (2005) referencing technique.

In addition to the logarithmic receiver, the EarthCARE CPR employs a linear detector for the estimate of the Doppler velocity (Battaglia and Kollias, 2014c). In the linear receiver, the analog signal is demodulated down to the baseband frequency prior to digital sampling by the Analog-to-Digital Converter (ADC). The resulting signal is usually referred to as complex demodulated, or I/Q-data, where I/Q stands for in-phase and quadrature-phase, reflecting the fact that the signal is complex, with a real and imaginary part.

In the EC-CPR receiver, a 1.5 MHz ADC sampling rate results to a range resolution of 100 m which implies a factor of 5 oversampling of the CPR true range resolution (500 m). The EC-CPR pulse repetition frequency (PRF) varies between 6.2 and 7.4 kHz. The return signal from each pulse results to another pair of I/Q at each range gate that includes contributions from the atmosphere (signal) and the radar receiver (noise). A new pair of I/Q samples is recorded every $\tau$ seconds, where $\tau = \frac{1}{PRF}$. The along track EC-CPR signal integration is 500 m. This implies that all the I/Q samples collected every 500 m





of along track satellite displacement are used to estimate the CPR Doppler radar moments. Using a reference satellite velocity $V_{sat}$ of 7.6 kms$^{-1}$, this results from 400 to 486 pairs of I/Qs every 500 m of along track integration for each sampling range gate depending on the PRF.

Within the 500 m along track integration, the I/Q samples are not recorded continuously. The CPR on-board processing unit uses 21-22 consecutive I/Q pairs at each CPR range gate (r, every 100 m) to provide an estimate of the autocovariance $R(r,0)$

and $R(r,\tau)$ at lag-zero and lag-one of the radar complex signal $V(r,t) = I(r,t) + jQ(r,t)$. Depending on the EC-CPR PRF, it takes 22-27 m of along-track displacement of the CPR to collect $M = 22$ consecutive pairs of I/Q. Next, the CPR receiver noise is measured during a period where the CPR does not transmit. The time spent to measure the radar receiver noise is the equivalent of 2 pulses. Thus, in total we have 24 pulses, 22 pulses for the estimation of $R(r,\tau)$ and the time for 2 pulses for the estimation of the radar receiver noise.

$$R(r,\tau) \equiv \frac{1}{M} \sum_{i=1}^{M} V(r,t)\, V^{\star}(r,t+\tau) \tag{1}$$

This process is repeated 17-20 times (depending on the PRF) within the 500 m along track integration. Every 500-m along track integration, the mean values of the $R(0)$ and $R(\tau)$ estimates are reported in the JAXA CPR L1b data product along with the raw estimate of the mean Doppler velocity $V_D$, and the spectrum width, $\sigma_D$, using the following expressions:

$$V_D(r) \;=\; \frac{\lambda}{4\pi\tau} \arctan\left\{ \frac{\mathcal{I}[R(r,\tau)]}{\mathcal{R}[R(r,\tau)]} \right\} \tag{2}$$

$$\sigma_D(r) \;=\; \frac{\lambda}{2\sqrt{2}\pi\tau} \sqrt{\left\{ 1 - \frac{R(r,\tau)}{R(r,0)} \right\}} \tag{3}$$

where $\mathcal{R}$ and $\mathcal{I}$ represent the real and imaginary components of a complex signal. In addition to the CPR primary measurements, the JAXA L1b CPR data product (called C-NOM) will include detailed geo-location information including the pitch, roll and yaw angle of the satellite, the velocity of the satellite along the flight direction, in the direction orthogonal to the orbit plane and the nadir direction.

## 75  3  CPR - Feature Mask and Reflectivity (C-FMR)

The C-FMR product output includes the feature (significant detection) 2-D (range, along track) mask based on only-reflectivity CPR measurements and quality-controlled radar reflectivity profiles. In addition to the standard geolocation variables, C-FMR contains the quality-controlled 94-GHz radar reflectivities both uncorrected and corrected for gaseous attenuation, estimates of total two-way gaseous attenuation as a function of along-track distance, the hydrometeor-induced path integrated attenuation

(PIA). Finally, the presence of multiple scattering (MS) in the EC-CPR observations is identified and appropriate flags are generated. The output of the C-FMR algorithm is provided at the Joint Standard Grid (JSG) resolution defined to bring together



the active and passive EarthCARE measurements. The vertical resolution of the JSG is 100 m (similar to that of the EC-CPR) and the along track resolution is 1000 m (twice the resolution of the raw EC-CPR measurements).

## 3.1 Feature Mask (FM) algorithm

One of the most important modules of the C-FMR product is the Feature Mask (FM) algorithm that identifies CPR returns that contain meteorological signal whose radar return power statistically exceed the background EC-CPR receiver noise and its fluctuation. The FM algorithm is based on Clothiaux et al. (1995); Marchand et al. (2008).

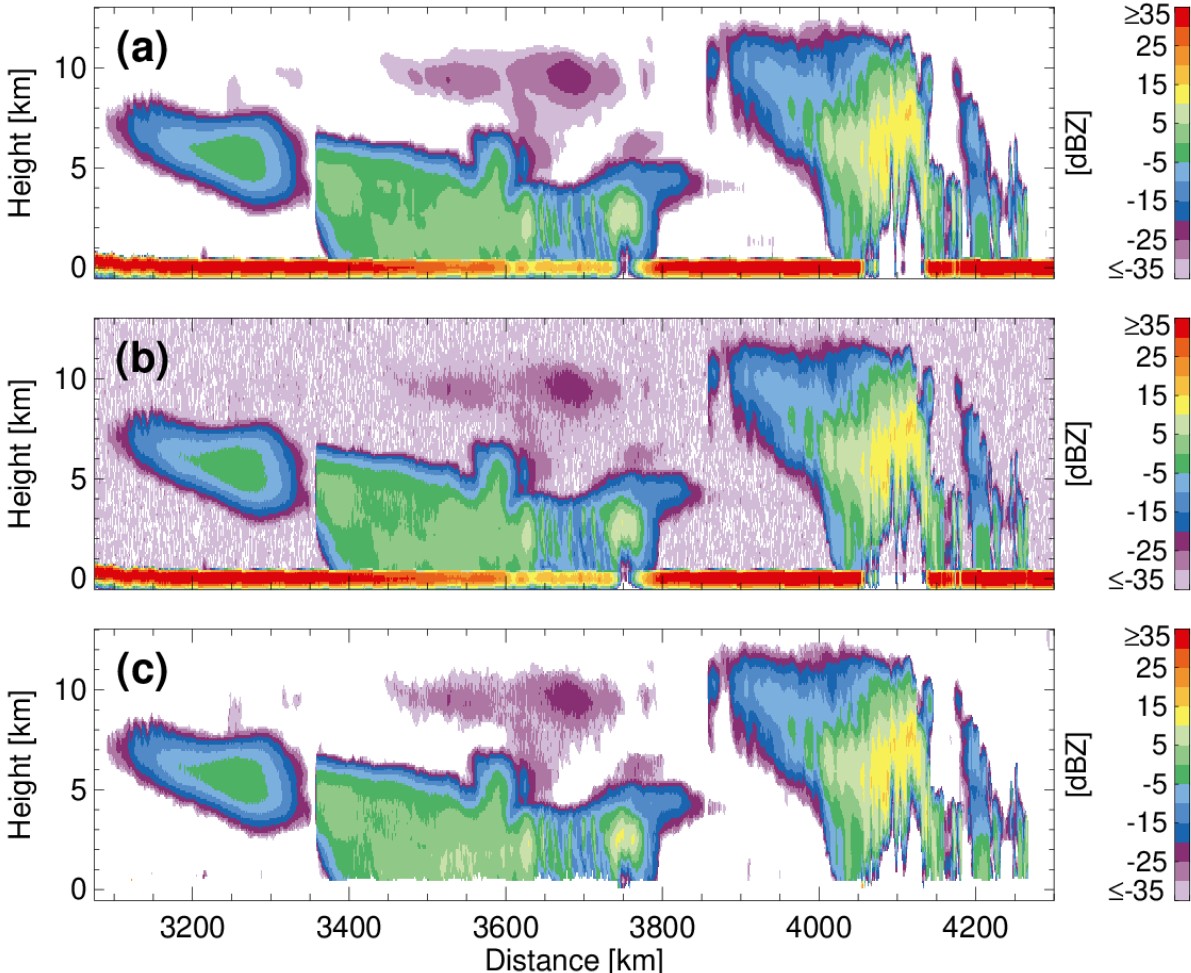

**Figure 1.** a) The truth hydrometeor locations in the Halifax scene as depicted by the radar reflectivity factor estimated from the GEM model output (at the radar resolution) with no radar receiver noise, b) the forward simulated CPR radar reflectivity (available in C-NOM) with radar receiver noise and c) the FM algorithm output.





In Fig. 1, panel (a) indicates the true hydrometeor locations based on the ECCC GEM model output for the Halifax scene. The hydrometeor locations are resampled from the GEM model resolution (3D model output with horizontal resolution of 250 m and vertical resolution of 100 m) to the EC-CPR resolution (100 m vertical resolution and 500 m along track resolution) using a sophisticated spaceborne radar simulator that accurately accounts for all the technical specifications (i.e., antenna pattern, range weighting function, along track integration) of the EC-CPR (Kollias et al. (2014, 2022)) . In addition to the ECCC GEM hydrometeor locations, panel (a) also includes the Earth's surface return and gaseous and hydrometeor attenuation at 94-GHz. This explains the missing hydrometeor locations in the low levels around 3780 - 3800 and 4050 - 4180 km due to total signal extinction.

Panel (b) shows the simulated output of the EC-CPR receiver as it will be available in the JAXA L1b CPR file (C-NOM). On average, half of the hydrometeor free space is occupied with signal + noise detections that exceed the average EC-CPR noise power. The FM algorithm objective is to remove these faint false "detections" while retaining as many as possible of the weak real detections.

Panel (c) indicates that the FM algorithm can identify most CPR significant detection's . In addition, the FM algorithm identifies and removes the surface clutter using a reference profile for the surface echo that is based either on existing (pre-launch) profiles for a given surface normalized cross section or using a clear-skies surface clutter profile if sufficient clear skies profiles are available "locally" (within 200 km and only over the ocean). A quantitative assessment of the performance of the CPR FM mask can be accomplished using the ECCC scenes and by characterizing the "hits", "misses" and "false detection's" of the FM mask. The overall equitable threat score (ETS) is 0.93 and the critical success index is 0.94.

## 3.2 Path Integrated Attenuation (PIA) Estimation

Neglecting multiple scattering effects (Battaglia et al., 2008; Battaglia and Simmer, 2008; Battaglia et al., 2010) the Path Integrating Attenuation (PIA in dB) is defined as the two-way, integrated extinction due to hydrometeors (Meneghini et al., 1983; Haynes et al., 2009):

$$PIA = 2\frac{10}{\log(10)}\left[\int\limits_{0}^{H} k_{ext}(z)\,dz\right] \tag{4}$$

where $k_{ext}$ is the extinction coefficient due to clouds and precipitation (Fig. 2). In the Rayleigh regime (hydrometeor diameter less than 800 $\mu$m at 94-GHz), the extinction is generally dominated by absorption. As absorption is proportional to the total liquid mass in the atmospheric column, PIA can be related to the total liquid mass in the atmospheric column (i.e. liquid water path (LWP)). In Fig. 2a, an example of strong 94-GHz signal extinction is evident at 2700 - 2750 km where the 94-GHz does not penetrate into a convective core. Spaceborne radars generally receive their strongest echoes from the Earth's surface. The radar echo from an extended surface is expressed in terms of the normalized (per unit of area) cross section of the surface, $\sigma_0$, (Hawkness-Smith, 2010). In spaceborne radars, PIA is estimated by measuring the depression of the measured surface echo $\sigma_0$ between cloudy and clear sky columns (local reference echo). If $\sigma_0$ is the normalized cross section of the surface and $\sigma_{clr}$ is



the clear sky cross section, then the hydrometeor PIA can be estimated as:

$$120 \quad PIA = \underbrace{(\sigma_{noatt} - A_g)}_{\sigma_{clr}} - \sigma_0 \qquad (5)$$

where $\sigma_{noatt}$ is the unattenuated ocean surface normalized cross section estimated using the relationship from Li et al. (2005) as a function of the near surface wind speed provided in the X-MET data product. $A_g$ is the gaseous attenuation estimated using the Rosenkranz (1998) absorption model and the X-Met provided temperature and moisture profiles matched to the observations of the spaceborne radar. CloudSat observations have shown that over the ocean surface $\sigma_0$ is known within 2 dB and over land exhibits very large variability due to its dependency on vegetation, surface slope, soil moisture, snow cover and other factors (Haynes et al., 2009). Thus, the estimation of PIA is only possible over the ocean.

The surface normalized cross section is estimated by integrating the surface echo return at CPR ranges ±500 m (Fig. 2). The CPR 100-m range resolution (compared to the 240 m for CloudSat) improves the integration of the surface echo return and the $\sigma_0$ estimation. However, the estimation of the $\sigma_{noatt}$ is sensitive to the accuracy of the near surface winds. In the ECCC simulations, the surface wind conditions are well known, thus, the accuracy of the retrieved PIA estimates are overly optimistic (Fig. 2b).

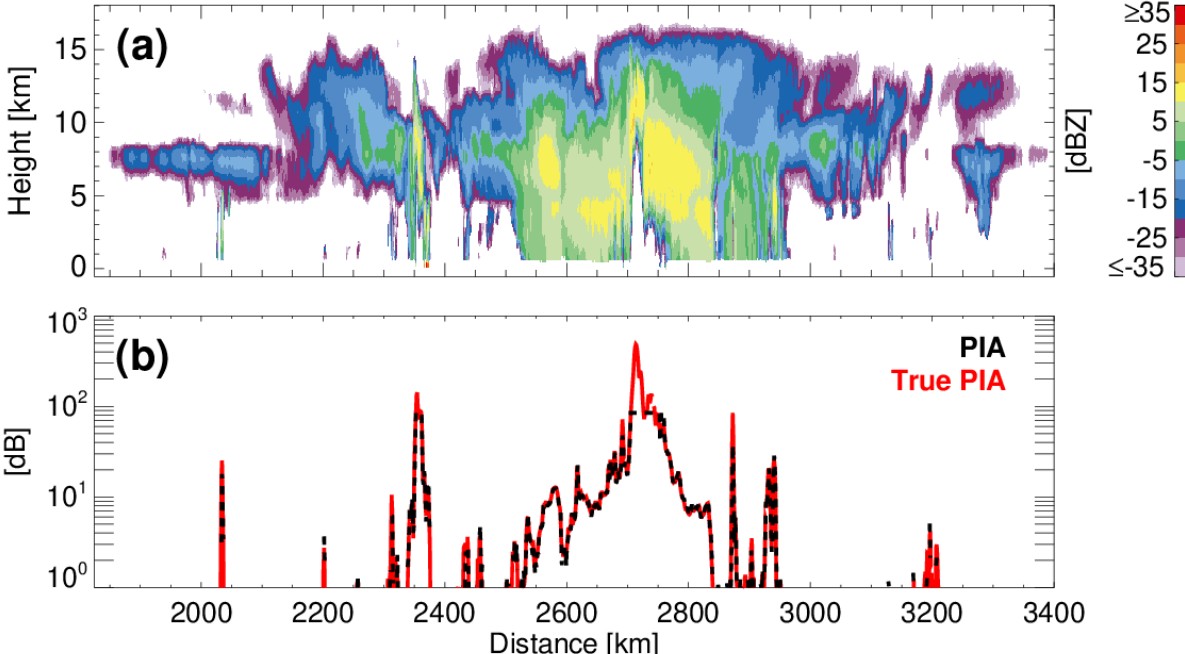

**Figure 2.** a) The CPR radar reflectivity from the 1825 - 3400 km along-track segment of the Hawaii (tropical) scene, b) the C-PRO estimated (black like) and true (red line) PIA time series from the same segment of the Hawaii scene over the ocean.





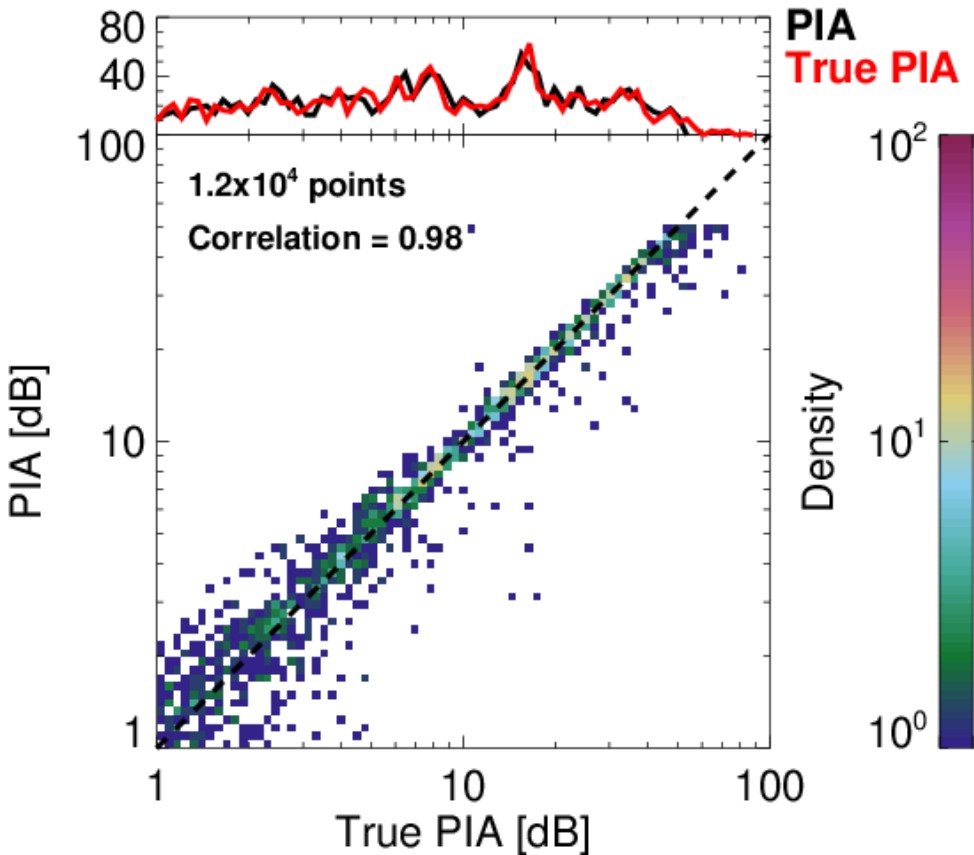

**Figure 3.** Scatter plot of the estimated C-PRO PIA and the true PIA from the three scenes from ECCC

Fig. 3 shows the comparison of the true and estimated PIA from the three ECCC model scenes. The agreement is very good since the only source of error in this comparison is due to the introduction of noise in the radar measurements and the uncertainly introduced by estimating $\sigma_{noatt}$ using the EC-CPR measurements around the surface range gate. The uncertainty in the $\sigma_{noatt}$ estimation using the Li et al. (2005) methodology can be as high as 0.5-1 dB (Haynes et al., 2009; Battaglia et al., 2020a). This suggests that the PIA can be a useful constraint in precipitation retrievals when the precipitation layer is deep (more than 1 km) and for rainfall rates higher than 1-2 mmh$^{-1}$. However, in the case of lighter precipitation (drizzle) or liquid clouds, a more robust method for the estimation of $\sigma_{clr}$ is desirable. In C-FMR, we apply (when possible) the local reference technique proposed by Hawkness-Smith (2010). The local reference technique is based on the suggestion that the absolute value of $\sigma_{noatt}$ is not important since it is the difference (depression) of the surface normalized cross section $\sigma_{clr} - \sigma_0$ that determines the PIA. Thus, by interpolating clear skies values of $\sigma_{clr}$ embedded in cloudy scenes an improved estimate of the PIA can be retrieved in cloudy scenes. The first step is the determination of clear skies scenes. Based on the feature mask





output, a clear-skies column is defined as one that contains no CPR range gates with hydrometeor detection. If the along-track length of the clear-skies region is less than 5 km, then the $\sigma_{clr}$ is estimated as the average value of $\sigma_{clr}$ for this region. For

longer along-track extents, a 5-km long running mean window is used to estimate the $\sigma_{clr}$ values. The clear-skies $\sigma_{clr}$ values are interpolated in cloudy along-track regions to provide $\sigma_{clr}$ in cloudy regions. If the along-track spacing between the clear regions is less than 250 km, then the interpolated values are used. If the along-track spacing between the clear regions is more than 250 km, then the relationship from Li et al. (2005) is used.

### 3.3 Multiple scattering (MS) Detection

For spaceborne millimeter wavelength radars MS and attenuation are two different manifestations of the same underlying phenomenon, i.e. the multiple interaction of the emitted radiation within the radar field of view (Battaglia et al., 2010). In the CloudSat CPR observations MS is ubiquitous particularly in the presence of deep convection where higher ice contents and denser ice particles are more likely to occur. Fig. 4a,b show an example of CPR simulated observations from a deep convective tropical system (Hawaii ECCC model scene) using single scattering (Fig. 4a) and multiple scattering (Fig. 4b). The stretched

MS echoes are clearly visible in the low-levels of the convective core (2700 - 2760 km). These echoes are not real and should be flagged as MS echoes.

In the C-PRO the Battaglia et al. (2011) criterion for detecting MS in W-band spaceborne radar observations is applied. For this, the integral of the radar reflectivity above a certain threshold value ($Z_{thres}$) from the top of the atmosphere (TOA) down to a level $z$ is computed at each CPR profile:

$$I(z) = 10\log_{10}\left[\int_z^{TOA}\left\{Z_{OBS} - Z_{thres}\right\}(z)\,dz\right] \qquad (6)$$

where the integral is performed only at those heights where the CPR radar reflectivity $Z_{OBS}$ exceed in magnitude a threshold value $Z_{thres}$. Following Battaglia et al. (2011), for the EarthCARE CPR technical specifications, the best statistical match for convective profiles is achieved when $Z_{thres}$ is selected equal to 12 dBZ. MS is likely to be encountered below the height $z$ where $I(z)$ exceeds 41 dBZint (dBZint is the unit when the integral in Eq. (6) is in mm$^6$/m$^2$). Below this height all CPR

observations are flagged as containing significant MS contributions (Fig. 4c).

### 4 CPR - Corrected Doppler (C-CD)

The estimation of the EC-CPR Doppler velocity is complicated due to the considerable platform motion ($V_{sat}$ = 7.6 km s$^{-1}$). The EC-CPR transmits a 3.3 $\mu$m pulse from a single antenna. In this configuration, the EC-CPR Doppler velocity estimation is not based on polarization diversity techniques (Kobayashi et al., 2002; Battaglia et al., 2013; ?) or the Displaced Phase

Center Antenna (DPCA), (Kollias et al., 2022) concept that can minimize the impact of the high platform motion. Considering that the antenna (and thus the antenna beamwidth) are fixed, the only remaining parameter that controls the performance depends considerably on the selected PRF (Kobayashi et al., 2002) which varies between 6100 and 7400 Hz in an orbit. The







**Figure 4.** The C-NOM reflectivity simulations for a) single and b) multiple scattering and c) the C-PRO multiple scattering detection mask where SS stands for Single Scattering and MS for Multiple Scattering

high platform speed introduces significant signal decorrelation from pulse-to-pulse (Battaglia and Kollias, 2014a) and it is manifested as broadening of the radar Doppler spectrum (3.6 - 3.8 m$s^{-1}$ for the EC-CPR). The aforementioned broadening

is significant if we consider that the EC-CPR Nyquist velocity $V_N$ is between 5 and 6 m$s^{-1}$ (Tanelli et al., 2002; Kollias et al., 2014; Illingworth et al., 2015; Kollias et al., 2022). The result of this broadening is a significant increase in the EC-CPR Doppler velocity measurement uncertainty especially at low Signal-to-Noise (SNR) conditions. If the distribution of the targets within the EC-CPR sampling volume is uniform, then the broadening increases the uncertainty but introduces no Doppler velocity bias. However, if the EC-CPR sampling volume is characterized by non-uniform beam filling (NUBF) conditions

especially in the along-track direction, then in addition to the broadening, we have a Doppler velocity bias (check Doppler velocity explanation box in (Illingworth et al., 2015)). The NUBF-induced Doppler velocity bias is proportional to the square





of the length of the EC-CPR Instantaneous Field of View (IFOV) and the along track gradient of the radar reflectivity within the EC-CPR sampling volume (Battaglia et al., 2020a; Kollias et al., 2022).

In addition, it is important to apply appropriate corrections to account for the EC-CPR antenna pointing off the geodetic nadir (Tanelli et al., 2005; Battaglia and Kollias, 2014b). The JAXA CPR L1b data product (C-NOM), includes satellite ancillary data with geolocation information provided by the satellite Attitude and Orbit Control System (AOCS). The spacecraft attitude is determined using a star tracker with a sampling rate of 20 Hz that translates to an rms in the knowledge of the EC-CPR antenna pointing $\Theta_{AOCS}$ of 10-15 $\mu$rad, which corresponds to an rms on the measured Doppler velocity of 0.08-0.11 ms$^{-1}$.

Another source of error is the Doppler velocity folding (aliasing) when the observed Doppler velocities exceed the Nyquist velocity $V_N$. Depending on the EC-CPR PRF, the $V_N$ ranges from 5 to 6 m$s^{-1}$. The fall velocity of raindrops and the strength of convective dynamics (Kollias et al., 2018, 2022) suggest that there will be areas where velocity aliasing will take place. Several velocity unfolding algorithm exist for cloud radars (Kollias et al., 2014), however, in the case of the EC-CPR the large uncertainty in the Doppler velocity measurements can make the velocity unfolding challenging.

## 4.1 Doppler velocity corrections

### 4.1.1 Non-Uniform Beam Filling

The EC-CPR sampling volume has a vertical dimension of 500 m and a horizontal dimension of 750-800 m. Cloud and precipitation microphysics and dynamics can vary considerably within such atmospheric volumes. The 3D distribution of hydrometeors and turbulence will produce an inhomogeneous 3D field of radar reflectivity and Doppler velocities. At the end of each signal integration (500 m along track integration in the case of the EarthCARE CPR), the radar reports a single radar reflectivity and Doppler velocity. Thus, the radar sampling volume acts as a spatiotemporal low-pass filter and its impact on the desirable measurements should be considered (Kollias et al. (2022)). In addition to the low-pass filtering effects, for a spaceborne radar, the inhomogeneities in the radar reflectivity field $Z_e(x)$ especially in the along track direction (x) can introduce significant Doppler velocity biases. Tanelli et al. (2002); Kollias et al. (2022) have shown that the NUBF is a significant source of error in both time-domain and frequency-domain based estimates of Doppler velocity from spaceborne radars. Such issue could be mitigated by adopting large antennas that will reduce the radar footprint at the ground but this represents a technologically challenging and costly solution. Others configurations like displacement phase center antennas are currently under consideration (Durden et al., 2007; Battaglia et al., 2020a; Kollias et al., 2022)

Each point $x'$ in the along track direction within a spaceborne radar sampling volume at distance $h_{SAT}$ that moves with velocity $v_{SAT}$ has an apparent Doppler velocity, $V_{D,obs}$, that it is different from the true Doppler velocity, $V_{D,true}$, by the following expression:

$$V_{D,obs} = -\frac{v_{SAT}}{h_{SAT}} x' + V_{D,true}. \qquad (7)$$

Forward points ($x' > 0$) have an upward (towards the radar, negative sign) apparent Doppler velocity and aft points within the radar beam ($x' < 0$) have a downward (away from the radar, positive sign) apparent Doppler velocity. Their contributions





cancel out if their relative weights are equal. The weight of each point $x'$ is the product of its measured radar reflectivity $Z_e(x')$
and the antenna gain function $W_x(x')$. In NUBF conditions, $Z_e(x)$ is not symmetrical in the along-track direction and thus, the contributions from the forward and aft volumes of the EC-CPR beam do not cancel out, thus producing a Doppler velocity bias.

Tanelli et al. (2002); Sy et al. (2014) have demonstrated that the NUBF Doppler velocity biases correlate well with the gradient of the along-track radar reflectivity within the CPR sampling volume:

$$V_{D,true} = V_{D,obs} - \alpha \frac{\Delta Z_e}{\Delta x} \tag{8}$$

where $\alpha$ is the correlation coefficient in m s$^{-1}$/(dB km$^{-1}$) between the NUBF Doppler velocity bias and the along-track derivative of the measured reflectivity $Z_e$ expressed in dBZ. The reflectivity gradient is computed via a central finite-difference formula between consecutive samples. This implies that, given a 500-m sampling of $Z_e$, the derivative $\Delta Z_e/\Delta x$ is computed over a baseline of 1 km. Though various methods can be considered to determine $\alpha$, different studies have shown that a value
in the range between 0.17 and 0.23 m s$^{-1}$/(dB km$^{-1}$) generally produces the best performances in terms of bias reduction (Sy et al., 2014). Using the three ECCC scenes, the value of $\alpha$ used slightly depends on the magnitude of $\Delta Z_e/\Delta x$.

Note that the NUBF corrections are applied in the complex $R(\tau)$ lag-one of the radar complex signal $V(t) = I(t) + jQ(t)$. First, the $-\alpha \frac{\Delta Z_e}{\Delta x}$ is used to correct the phase $\phi_{D,obs}$ of the observed $R(\tau)$ for the rotation $\phi_{NUBF}$ induced by the NUBF conditions:

$$R_{Corr}(\tau) = R(\tau)e^{-j\phi_{NUBF}} = |R(\tau)|e^{j\phi_{D,obs}}e^{-j\phi_{NUBF}} = |R(\tau)|e^{j\phi_{Corr}} \tag{9}$$

where

$$\phi_{D,obs} = \arctan \frac{\mathcal{I}[R(\tau)]}{\mathcal{R}[R(\tau)]} \tag{10}$$

$$\phi_{NUBF} = \frac{4\pi}{\lambda PRF}\alpha\frac{\Delta Z_e}{\Delta x} \tag{11}$$

$$\phi_{Corr} = \phi_{D,obs} - \phi_{NUBF} \tag{12}$$

The real and imaginary parts of the corrected correlation function in Eq. 9 are used in the along-track integration of the CPR Doppler velocity.

### 4.1.2 Velocity Unfolding

The EarthCARE CPR PRF determines the highest sampled frequency. This is often called Nyquist or folding frequency (f$_N$ = PRF/2), which is half the sampling frequency of a discrete signal processing system. Using the radar wavelength ($\lambda$), the
folding frequency is converted to folding velocity or as often-called Nyquist velocity ($V_N = \lambda PRF/4$). The radar can correctly measure velocities within the interval of $\pm$ V$_N$. Velocity folding occurs whenever the phase shift detected between sequential radar pulses exceeds V$_N$. In general, the observed velocity values (folded or not) and their true values are related by:

$$V_T = V_O \pm \eta V_N \tag{13}$$





where $V_T$ denotes the true velocity, $V_O$ is the observed velocity by the radar and $\eta$ is an integer (0, 1, 2, . . .). The correction

of aliased velocities—the so-called dealiasing or unfolding—is a challenging technical task and becomes increasingly difficult with a decreasing Nyquist velocity or increasing noise in the data. Since aliasing is easily identified as abrupt changes in the velocity data field, most of the dealiasing techniques are based on detecting spatial and temporal discontinuities. In profiling radars, Doppler velocity folding occurs due to presence of either fast falling hydrometeors and/or strong dynamical drafts in the radar resolution volume. These dynamical and microphysical effects exhibit coherency in time-height and can be identified

and corrected if a reference velocity is available somewhere in the profile (e.g., cloud top, or at low radar reflectivity values).

However, in the case of EarthCARE, the application of this approach is not straightforward. The CPR Doppler velocities are characterized by large uncertainties that can lead to aliasing in the absence of microphysical and/or dynamical effects. Furthermore, NUBF conditions can also lead to velocity aliasing in the absence of microphysical and/or dynamical effects. The most challenging scenario for applying the "reference velocity" technique is in convective clouds, due to their strong

vertical air motion variability and the presence of strong NUBF. For those reasons, the velocity-unfolding algorithm applied to the EC-CPR is only reliable for cloud and precipitation systems characterized by weak dynamics with vertical air motion $|w_{air}| < 2 \text{ ms}^{-1}$. In such stratiform conditions, the EC-CPR Doppler velocity can fold only around its positive limit ($+V_N$) and the correction is straightforward since $\eta = 1$.

### 4.1.3 Spatial averaging

After the implementation of the aforementioned corrections, the EC-CPR Doppler velocity estimates at 500-1000 m along track resolution are still characterized by large uncertainty. The large uncertainty in the Doppler velocity measurements are associated with the decorrelation of the signal due to Doppler fading and low Signal-to-Noise conditions (Kobayashi et al. (2002); Kollias et al. (2014)). The only remaining technique to reduce the uncertainty in the EC-CPR Doppler velocity measurements is the implementation of spatial averaging (in the along track and vertical dimensions). At 500 m along track resolution, the EC-CPR

Doppler velocity uncertainty is approximately 1 $\text{ms}^{-1}$. A 5 km along track averaging should reduce the EC-CPR Doppler velocity uncertainty to $< 0.3 - 0.4 \text{ ms}^{-1}$ (Kollias et al., 2022). This expected reduction in the EC-CPR EC-CPR Doppler velocity uncertainty will facilitate the proper interpretation and use of the Doppler velocity measurements in downstream microphysical algorithms such as the C-CLD and ACM-CAP (Mroz et al., 2022; Mason et al., 2022). However, the spatial averaging of the EC-CPR Doppler velocities comes at the cost of a coarser spatial sampling of the final radar product, which

raises the issue of the representativeness and practical usefulness of the integrated data (Sy et al. (2014); Kollias et al. (2014)). The CPR radar reflectivity is the only piece of information that can be used to describe the scene microphysical variability at any given range gate. Particle sedimentation regimes (ice clouds, drizzle, stratiform precipitation) are generally characterized by gentle gradients of radar reflectivity.

In the C-CD data product, a 2-D integration window with an along-track length $L_x$ (km) and a vertical length $L_z$ (km) is

introduced. The averaging is conducted using the $R(\tau)$ estimates within the window. The sizes $L_x$ and $L_z$ of the integration window can be generally scene dependent. The spatial filtering is estimated using the following procedure:



1. the length $L_x$ of the integration window is set to 5 km;

2. the length $L_z$ of the integration window is set to 300 m;

3. the integration window should not include CPR detection with reflectivities lower than -20 dBZ.

4. the integration window should not include CPR detections with multiple scattering flag;

5. the edge of the integration window should be at least 1 km away from a lateral cloud/precipitation boundary based on the CPR feature mask.

Once the integration window is determined, first the average $\mathcal{R}[\langle R \rangle_{L_x, L_z}(\tau)]$ and $\mathcal{I}[\langle R \rangle_{L_x, L_z}(\tau)]$ are estimated using the high resolution (500-m) along track measurements of $\mathcal{R}[R(\tau)]$ and $\mathcal{I}[R(\tau)]$ and then along track integrated velocity is estimated.

## 4.2 Sedimentation Velocity Best Estimate (SVBE)

One of the primary scientific objectives of the EC-CPR is the characterization of the global climatology of hydrometeors sedimentation (fall) velocity over a wide range of meteorological and aerosol conditions (Illingworth et al., 2015; Kollias et al., 2022). Fig.5a shows an example of the reflectivity-weighted hydrometeor sedimentation Doppler velocity. At the upper part of the widespread precipitating system particles sediment slowly. At warm temperatures the microphysical processes of aggregation and riming contribute to the increase of their sedimentation velocity. Finally, at the $0^o C$ isotherm, the melting of the solid hydrometeors to raindrops further increases their sedimentation velocity. Fig. 5b shows the corresponding raw, uncorrected EC-CPR Doppler velocities. The only correction that has been applied in Fig. 5b is the antenna pointing correction. Except the area at 2700 - 2780 km along range that is characterized by strong convection, the remaining area is characterized by very weak dynamics. As a result, the raw, uncorrected field of EC-CPR Doppler velocities albeit noisy, resembles the true hydrometeor sedimentation velocity. This uncertainty in the CPR mean Doppler velocity is too high and will hinder our ability to constraint the hydrometeor size information (e.g., median volume diameter $D_m$ estimation in C-CLD) especially in light precipitation (drizzle) and ice/snow sedimentation regimes. The application of the spatial averaging (Fig. 5c) substantially reduces the EC-CPR Doppler velocity uncertainty but still increase areas where the EC-CPR is negative suggesting that vertical air motion and remaining uncertainty in the EC-CPR Doppler velocities affects the overall sign of the EC-CPR Doppler velocity.

A profiling radar does not directly measure the hydrometeor sedimentation velocity. In principle, the observed Doppler velocity, $V_D$, from a profiling (nadir or zenith pointing) radar is the sum of the hydrometeor fall velocity (weighted by the backscattering cross section and the number concentration), $V_F$, and the vertical air motion, $V_A$:

$$V_D = V_F + V_A. \tag{14}$$

The relative contribution of the two in $V_D$ depends strongly on the convective nature of the clouds and the size of the hydrometeors (e.g., the radar reflectivity). Thus, it is important that we identify the hydrometeor type and the dynamical state



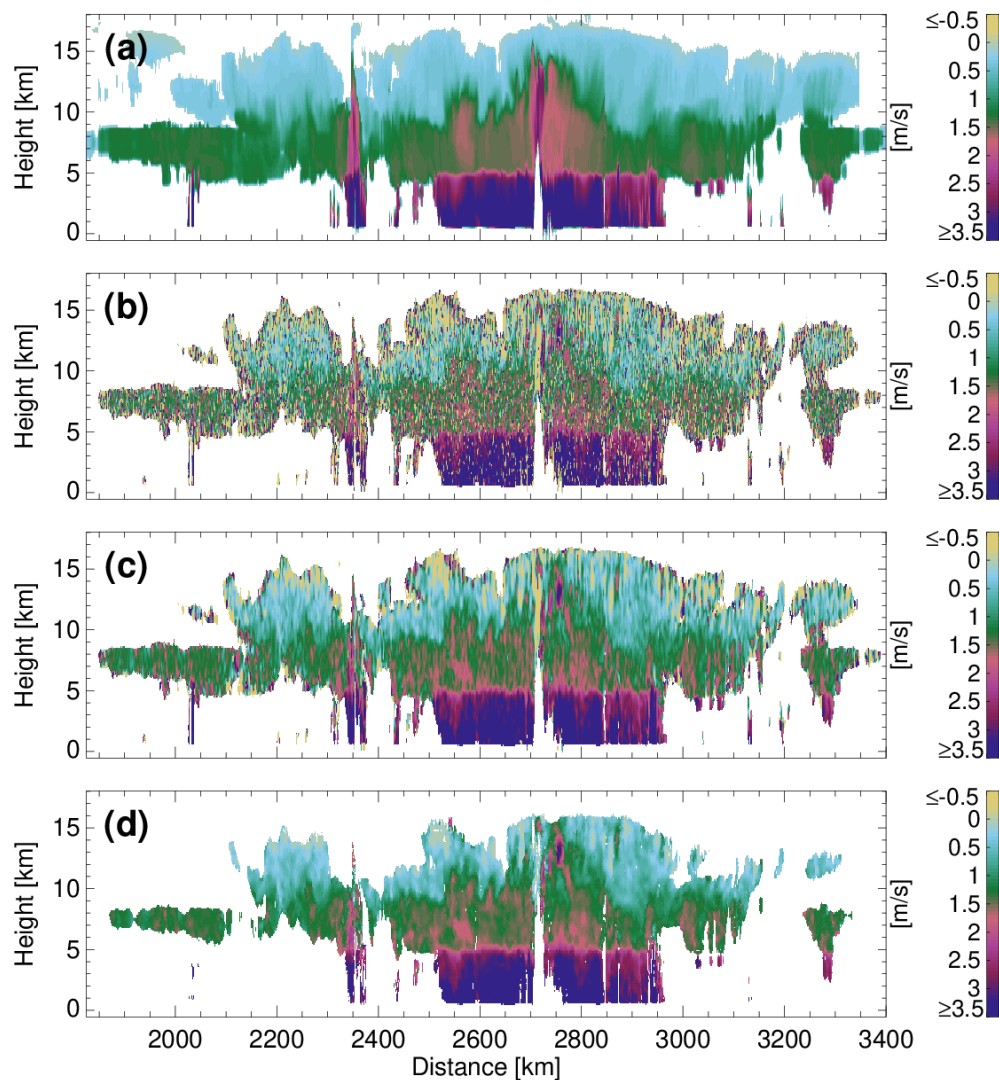

**Figure 5.** a) The true hydrometeor sedimentation velocity from the GEM model for the Hawaii scene, b) the raw, uncorrected CPR Doppler velocities, c) the 4-km along track and 500 m in the vertical integrated Doppler velocity and d) the sedimentation velocity best estimate (SVBE).

of the cloud/precipitation scheme before we interpret the observed Doppler velocities from space. In C-CD, a hydrometeor sedimentation velocity best estimate (SVBE) is inferred. The SVBE is used as input to single instrument (C-CLD, Mroz et al. (2022)) and synergistic (ACM-CAP, Mason et al. (2022)) microphysical algorithms and provides constraints on particle size and density.

The SVBE estimation is achieved by averaging radar observations within a narrow range of radar reflectivity and at different heights (Kalesse and Kollias, 2013) using a methodology similar to the $V_t - Z_e - H$ technique, described in Protat and Williams






(2011) (Fig. 5d). The algorithm is progressively applied at four different along track windows [40, 30, 20 and 10 km] starting
with the largest window. The vertical dimension of the along track windows is 3 CPR range gates (300 m). At each window,
reflectivity bins are defined, ranging from -15 to 20 dBZ every 3 dB. The minimum of -15 dBZ has been determined using
numerical simulations that indicate that the EC CPR Doppler velocity measurements are reliable only for SNR values exceeding
+6 dB. The single-pulse sensitivity (SNR = 0) of the EC-CPR is close to -21 dBZ, thus, a -15 dBZ value corresponds to an
SNR value of +6 dB. As a result, the SVBE algorithm does not assign sedimentation velocity for CPR reflectivity value below
-15 dBZ.

If the number of CPR Doppler velocities within a particular radar reflectivity bin exceed a minimum threshold (5), then
the CPR Doppler velocities within the same radar reflectivity bin are averaged. The assumption here is that the averaging
will remove or minimize the vertical air motion contribution assuming that there is no correlation between the hydrometeors
reflectivity and vertical air motion. The averaged velocity within every radar reflectivity bin is the SVBE for all the CPR
observations in the window that have values that fit within the particular CPR reflectivity bin. The process is repeated for all
radar reflectivity bins and for all different along-track windows. At the smaller windows, the probability of finding 5 CPR
values within a particular reflectivity bin decreases. However, when available, the SVBE estimates from smaller windows are
preferred as they represent better spatial microphysical inhomogeneities. If the SVBE estimates at the smaller window are
not available, then they are replace by those provided by a larger window applied in the same area of CPR observations. All
four along track windows are applied to CPR observations with no overlap in the vertical but they overlap by 50% in the
along track direction. Fig. 5d shows an example of SVBE values. The SVBE estimates are always positive. This is consistent
with the expected sign of sedimentation velocities as shown in Fig. 5a and this facilities the direct import of the SVBE into
microphysical retrievals. Furthermore, there are no SVBE estimates near the cloud edges due to sampling size issues and near
the cloud top where the CPR reflectivities are below -15 dBZ.

A summary of the performance of the different Doppler velocity estimates is provided in Fig. 6. Each estimate is compare
against its true value from the three ECCC model scenes resampled at the CPR resolution using our CPR instrument simulator.
The Root Mean Square Error (RMSE) is plotted as a function of the true hydrometeors sedimentation velocity. The RMSE of
corrected for antenna pointing only EC CPR Doppler velocities at 500 m along track resolution is shown with the purple line
in Fig. 6b. The uncertainty is approximately 1.5 m$s^{-1}$. At sedimentation velocity values large than 3 m$s^{-1}$ the RMSE value
increases significantly due to velocity folding that is not corrected here. It is also important to note that more than 90% of the
data points have true sedimentation velocities below 2.5 m$s^{-1}$ Fig. 6a. The application of the NUBF correction (blue line in
Fig. 6b) results only to a small reduction in the RMSE value. This is attributed to: i) the narrow IFOV of the EC-CPR (750-800
m) that controls the magnitude of the NUBF Doppler velocity bias (Kollias et al., 2022) and ii) the small fraction of convective
conditions with appreciable values of along-track gradient of the radar reflectivity.

As expected, the application of the spatial filtering (or along-track integration) has the largest impact in terms of RMSE
reduction (green line). The RMSE value is around 0.5 m$s^{-1}$ for the majority of the observations (below 1.8 m$s^{-1}$). The
application of the SVBE technique further reduces the RMSE with a value close to 0.3 m$s^{-1}$ in the same range of sedimentation
velocities. The RMSE values (Fig. 6b) indicate that we can estimate the SVBE with an uncertainty of 0.3-0.4 m$s^{-1}$ around 80



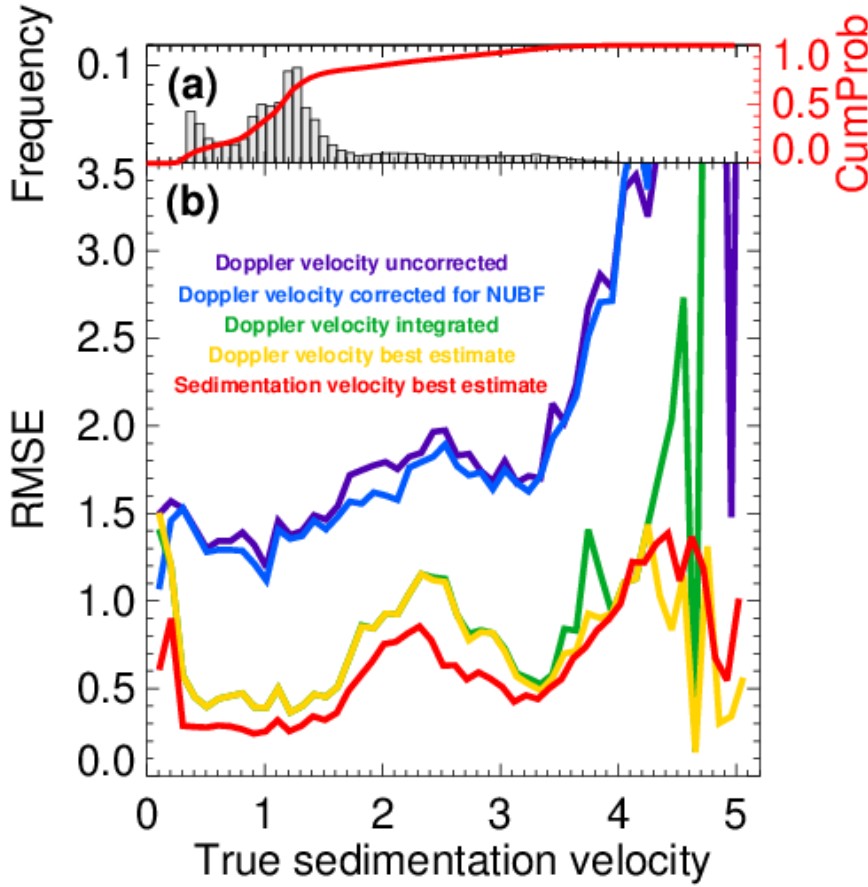

**Figure 6.** a) The frequency of occurrence and the cumulative distribution of occurrence of Doppler velocities in the three ECCC scenes, b) the Root Mean Square Error (RMSE) in Doppler velocity estimation as a function of the magnitude of the true sedimentation velocity for the three GEM scenes.

percent of the time (see cumulative probability in Fig. 6a). The noticeable increase in the RMSE between 1.7 and 3.2 ms$^{-1}$

(Fig. 6b) is caused by areas in the ECCC models with a lot of variability, presence of strong updrafts and graupel.

## 5 CPR - Antenna Pointing Characterization (C-APC)

The CPR Antenna Pointing Correction (C-APC) processor i) applies the antenna pointing correction based on the Attitude and Orbit Control System (AOCS) data and ii) investigates any additional CPR antenna pointing miscalibration that is not captured by the AOCS. Possible sources of error in the reported CPR antenna pointing are technical challenges with the star

tracker sampling and the AOCS and thermoelastic distortions of the platform and instrument. The main input to the C-APC





processing algorithm is L1b CPR data (C-NOM). The first correction (Fig. 7A1) is straightforward and the corrected for AOCS EC-CPR Doppler velocities are used as input to a series of corrections to remove additional sources of biases. These corrections (Fig. 7A2-4) are applied on the CPR observations that come from two different natural targets: Earth's surface (Tanelli et al., 2005; Battaglia and Kollias, 2014c) and ice clouds (Battaglia and Kollias, 2014b). Intrinsic properties of natural targets are

commonly used to provide supplemental monitoring of radars: for instance, the differential reflectivity, $Z_{DR}$, of drizzle is used to set $Z_{DR}$ to zero when calibrating ground-based polarimetric radars or the ocean surface echo at $10°$ incidence angle can be exploited to calibrate the CPR radar reflectivity values.

An overall flow chart of the C-APC processing algorithms is shown in Fig. 7. The Earth's surface referencing technique works instantaneously (required only local observations, i.e. CPR observations within 5–20 km). Basically, in the absence of NUBF induced either from the variability of atmospheric path within the CPR

footprint or from the heterogeneity of the Earth's surface, the pointing induced bias is given by the CPR Doppler velocity of the surface echo. The heterogeneity of the surface within the CPR footprint (800 m) is expected to be a factor over land, thus, this technique is not recommended for application over land surface. Prior to using the ocean surface reference technique, the ocean surface raw EC-CPR Doppler velocities are corrected for NUBF bias (Fig. 7A2).

In addition to the Earth's surface, (Battaglia and Kollias, 2014b) demonstrated that the Doppler velocity in ice clouds can

be an excellent, alternative source for evaluating the pointing of the EC-CPR for two reasons : 1) they are ubiquitous over the planet, with good probability of occurrence at all latitudes over land and ocean and over all seasons; 2) the global distribution of ice clouds radar reflectivity weighted mean Doppler velocity is well known by ground-based radar measurements (e.g. Kalesse and Kollias (2013)) as a function of their radar reflectivity and or temperature.

An overall flow chart of the C-APC processing algorithms is shown in Fig. 7. The C-APC required input data are: the

JAXA CPR L1b C-NOM and the X-MET files. Specifically, the surface echo, land/water mask and temperature are used to identify the surface and ice clouds, respectively. The pitch angles reported by the AOCS will be also used to assess the mispointing uncertainties. Corrected reflectivities will also help in screening out low-quality calibration points (e.g. where surface reflectivity is highly variable or for ice clouds with low SNR).

Prior to using the ice clouds for evaluating the pointing of the EC-CPR, two corrections are applied. First, the raw Doppler

velocities are corrected for any NUBF-induced Doppler velocity bias (see Sect. 4). Second, using the Kalesse and Kollias (2013)) relationship between radar reflectivity and mean Doppler velocity for ice clouds, the ice clouds fall velocity that corresponds to a particular CPR ice cloud reflectivity is removed (Fig. 7A3). Next, the spatial filtering described in Sect. 4.1.3 is applied in the segment of the CPR observations that correspond to these two natural targets (Fig. 7A4).

One difference between the ocean surface and ice clouds referencing techniques is that the former can be applied locally (it

requires a minimum of 20-50 km of along-track ocean surface CPR Doppler velocity measurements) while the latter performs better if ice clouds CPR observations are available from a large segment of an orbit or even multiple orbits.

In the case of the ocean's surface technique, the departure of the filtered, quality-controlled ocean's surface CPR Doppler velocity from zero is converted to an antenna mispointing angle $\theta_{mp}$ that was not characterized by the AOCS. A low-pass harmonic function is fitted to the estimated $\theta_{mp}$ to further remove outliers and provide a relationship that describes the CPR



# C-APC Flowchart

**A1** — Correct ALL raw radar data for antenna pointing reported by the AOCS

**A2** — Correct Earth's surface raw radar data for NUBF

**A3** — Correct Ice Clouds raw radar data for NUBF and fall velocity bias

**A4** — Along-track integration of corrected raw radar data

**A5** — Produce Best Estimate of Antenna Pointing Characterization

**Figure 7.** The C-APC algorithm flowchart.





antenna pointing. As in the case of the NUBF (Sect. 4.1.1) the correction is applied to $R(\tau)$, the complex correlation at lag-one of the radar complex signal $V(t) = I(t) + jQ(t)$.

In the case of the ice clouds technique, due to the natural variability of the ice microphysics, the uncertainty in the relationship of the ice cloud fall velocity as a function of reflectivity and the presence of gravity waves, a localized determination of the antenna mispointing angle $\theta_{mp}$ is not recommended. The requirement for a large segment of CPR observations complicates the implementation of this method in the standard ESA ground-based L2 data product processing chain since it is based on the idea that each 1/8 of an orbit long data files can be autonomously processed to produce L2a/b and L3 products. In addition to the requirement for large segment of CPR observations, there is a need for ice clouds observations in the data segment. The C-APC data product is designed to ingest 1 full orbit (8 frames) of L1b CPR data but is also able to use the Earth's surface reference technique on a frame-to-frame basis.

In order to evaluate the performance of the two different referencing techniques, the three ECCC scenes have been modified and concatenated in order to simulate a full EarthCARE orbit (Fig. 8). With this complete synthetic orbit, a C-NOM file has been generated using the specifications of EarthCARE and following the C-NOM product definition. The CPR antenna mispointing is simulated using the methodology suggested in Battaglia and Kollias (2014b) and the resulting Doppler velocity bias is shown in Fig. 8a. The generated synthetic C-NOM and X-MET files have been ingested to C-APC, testing the performance of the two proposed reference techniques in recovering the harmonic behavior of the CPR antenna mispointing.

Fig. 8b shows the Earth's surface CPR Doppler velocities (black dots, available only over the ocean surface). The CPR Doppler velocities are influenced by the introduced CPR antenna mispointing, NUBF and the inherit Doppler velocity uncertainty due the platform motion. The Earth's surface observations are used to fit a harmonic function (red line) that correlates very well with the mispointing velocity introduced in the test data (Fig. 8a). The regression fit is considerably good with a coefficient of determination $r^2 = 0.91$.

Fig. 8c shows the ice clouds CPR Doppler velocities (black dots, available only when ice clouds are available). As expected the ice clouds referencing Doppler velocities are more noisy (Fig. 8c) and observations available from any particular frame will not be sufficient to retrieve the parameters of the simulated antenna mispointing. If all the frames of an orbit are available (8 frames per orbit), then the retrieved antenna mispointing correlates reasonably with the mispointing velocity data but the quality of the model fit (regression + polynomial fit of the residuals) is not as good, $r^2 = 0.53$.

Finally, Fig. 8d indicates the Doppler velocity residual after the C-APC algorithm is applied. When using the Earth's surface based antenna pointing characterization, the residual Doppler velocity exhibits an unbiased sinusoidal structure with an amplitude of 0.03-0.05 m$^{-1}$. When using the ice clouds based antenna pointing characterization, the residual Doppler velocity exhibits a bias of 0.1 ms$^{-1}$. The bias is due to the difference between the climatological $V_t - Z_{ice}$ relationship used in the C-APC algorithm and the actual $V_t - Z_{ice}$ relationship in the ECCC forward radar simulations that depends on the ECCC model ice particles mass, density and terminal velocity assumptions. Post EarthCARE launch, the comparison between the Earth's surface based and ice clouds based techniques will allow us to adjust the $V_t - Z_{ice}$ relationship used in the C-APC algorithm.

These results suggest that the Earth's surface correction technique works and it can be used to calibrate the EarthCARE mispointing angle. The ice clouds correction introduces more variability due to the uncertainty in the $V_t - Z_{ice}$ relationship.



**Figure 8.** a) Mispointing velocity introduced in the test data, (b) Earth's surface correction, (c) ice clouds correction and (d) mispointing velocity residuals of the Earth's surface, ice clouds and combination of Earth's surface and ice clouds correction together (for comparative purposes). The red, blue and black solid lines represent the regression fit.

Further analysis will be required to understand the limitations of the ice clouds velocity to reflectivity relationship. Post launch EarthCARE measurements will help determine the actual attitude of the antenna mispointing angle $\theta_{mp}$ and therefore, improve the technique.



## 6   Conclusions

The Earth Clouds, Aerosols and Radiation (EarthCARE) satellite mission is scheduled for launch in 2024. The EarthCARE
CPR will be the most sensitive radar ever in orbit. Due to its higher sensitivity and smaller footprint, the EarthCARE CPR is
expected to detect more non-precipitating clouds (Lamer et al., 2020) and provide improved estimates of shallow precipitation
(Battaglia et al., 2020b). In addition, the EarthCARE CPR will be the first atmospheric radar with Doppler capability in space.
The Doppler velocities from EarthCARE are expected to provide the first ever climatology of hydrometeors sedimentation
rates and improve microphysical retrievals (Kollias et al., 2022).

Here, the physical basis and algorithm structure of three of the CPR L2A algorithms is presented. The CPR feature mask
and reflectivity (C-FMR) product physical basis and algorithm structure is based on the strong heritage and experience gained
from NASA's CloudSat mission. The improved CPR receiver filter is expected to limit the impact of the Earth's surface echo
to 500 m above the ocean surface (Lamer et al., 2020) and in combination with the improved sensitivity is expected to lead to
more detections of low-level oceanic clouds (Burns et al., 2016). Unfortunately, the three ECCC scenes did not contain any
significant amounts of low-level oceanic clouds to allow us to test the performance of C-FMR algorithm under such conditions.

The other two CPR L2A products, the CPR Corrected Doppler (C-CD) measurements and the CPR Antenna Pointing
Characterization (C-APC) are targeting the quality control and interpretation of the first Doppler velocity measurements from
a spaceborne platform. A satellite platform is subject to less vibrations compared to an airborne platform (Heymsfield et al.,
2010), however, the higher platform motion introduces considerable uncertainty Doppler velocity estimates ($> 1ms^{-1}$) while
NUBF conditions and antenna mispointing can introduce Doppler velocity biases.

In the C-CD data product, the various steps used to mitigate some of the platform effects were described. Along track
integration has the largest improvement in terms of reducing the uncertainty of the EarthCARE CPR Doppler velocities. In
addition to reducing the uncertainty and removing biases in the EarthCARE CPR Doppler velocity measurements, the C-CD
data product introduces the SVBE, that provides the best estimate for the hydrometeors sedimentation velocity. The SVBE
estimates are reliable in cloud and precipitation systems characterized by stratiform conditions (i.e., weak vertical air motions).

In the C-APC data product, the various steps applied to mitigate any unknown amount of the CPR antenna mispointing are
described. Two natural targets are used to retrieve the amount of unknown CPR antenna mispointing: i) the Earth's surface and
ii) ice clouds. The former can be reliably used over the ocean surface and provide "localize" estimates of antenna mispointing.
The latter requires a larger data set (at minimum, it requires a significant fraction of a full orbit) of CPR Doppler velocity
measurements from ice clouds to capture the low-frequency behavior of the CPR antenna mispointing.

The presented algorithms and data products have been tested using synthetic observations from three ECCC model scenes
that cover a wide range of cloud and precipitation conditions and state of the art radar and orbit simulations that capture all the
known features of the instrument and of the satellite. The algorithms and data products will need to be revisited post launch for
revisions and adjustments once the real performance of the spacecraft and of the radar will be thoroughly characterised.



*Code and data availability.* The EarthCARE Level-2 demonstration products from simulated scenes, including the C-FMR, C-CD and C-APC products discussed in this paper, are available from https://doi.org/10.5281/zenodo.7117115 (van Zadelhoff et al., 2022)

*Author contributions.* All authors of this paper, namely Pavlos Kollias, Bernat Puigdomènech Treserras, Alessandro Battaglia, Paloma Borque and Aleksandra Tatarevic contributed fairly with regard to the development of the studies that led to the results presented here. They also contributed equally to the writing/correction of the different parts of the paper for which they are responsible.

*Competing interests.* At least one of the (co-)authors is a member of the editorial board of Atmospheric Measurement Techniques. The other co-authors have no other competing interests to declare.

*Acknowledgements.* Support for this work was provided by the European Space Agency (ESA) under the Clouds, Aerosol, Radiation – Development of INtegrated ALgorithms (CARDINAL) project (RFQ/3-17010/20/NL/AD).





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
