# Peer review of "Processing reflectivity and Doppler velocity from EarthCARE's cloud profiling radar: the C-FMR, C-CD and C-APC products"

_EGUsphere, 2022_

## Referee Comment (RC3)

Review of: **Processing reflectivity and Doppler velocity from EarthCARE's cloud profiling radar: the C-FMR, C-CD and C-APC products**, Pavlos Kollias et al.

Overall Recommendation: Publish with minor changes

Specific Comments:

- Line 21: Is it a good idea to mention a launch date within the paper?

- Line 52: Maybe mention already why this frequency varies and in which way (intentionally, technically,…).

- Line 70: A lot of a-priori knowledge is assumed. Estimation from autocovariance could actually be considered mathematical trivial, but a reference to a paper highlighting the technique would be really fruitful at this position. Actually, this technique only works if spectral leakage is avoided (i.e. the spectrum is fully and unambiguously recorded). So it should at least be shortly mentioned that it actually is applicable in the presence of all the adverse effects on the velocity measurement (NUBF etc...) and that it is representing the actual spectral width of the final velocity spectrum.

- Line 164: dBZint: please specify if this represents integrated Z values or the integrated linear values. I'm questioning if it is really necessary to define a new unit (dbZint) if a representation in actual physical units would be available.

- Figure 4: The classification of "Strong MS" is retrieved for areas where there is no signal in the ideal simulation (at around 2700km below 5km). It should be noted in the figure that it depicts the multiple scattering influence on the measured values and not the multiple scattering originating at the given location. This disambiguation is implicitly made in the text, but it should also be contained in the figure.

---

## Author Comment (AC1)

RC1: 'Comment on egusphere-2022-1284', Anonymous Referee #1

General comments

The manuscript of Kollias et al. describes the algorithms prepared for the level 2 processing of data to be obtained with the EarthCARE's Cloud Profiling Radar. It outlines the ideas used in the three algorithms, C-APC, C-FMR and C-CD. Results of performance tests of these algorithms with synthetic data sets are also shown. The paper is of use for users of EarthCARE's level 2 products. Although the manuscript is logically correct, it suffers from numerous minor editorial errors and awkward expressions.

This paper deserves ultimate publication; however, in order for it to be published, I would recommend that the manuscript be proofread by a native English speaker before publication.

The authors would like to thank the reviewer for their useful and insightful feedback. A point-by-point response to the reviewer's comments is provided below.

Evaluation

Does the paper address relevant scientific questions within the scope of AMT?  Yes.

Does the paper present novel concepts, ideas, tools, or data?  Yes.

Are substantial conclusions reached?  Yes.

Are the scientific methods and assumptions valid and clearly outlined?  Yes.

Are the results sufficient to support the interpretations and conclusions?  Yes.

Is the description of experiments and calculations sufficiently complete and precise to allow their reproduction by fellow scientists (traceability of results)?  Yes.

Do the authors give proper credit to related work and clearly indicate their own new/original contribution?  Yes.

Does the title clearly reflect the contents of the paper?  Yes.

Does the abstract provide a concise and complete summary?  Yes.

Is the overall presentation well structured and clear?  Marginally yes.

Is the language fluent and precise?  No.

Are mathematical formulae, symbols, abbreviations, and units correctly defined and used?  No.

Should any parts of the paper (text, formulae, figures, tables) be clarified, reduced, combined, or eliminated?  Yes.

Are the number and quality of references appropriate?  Yes.

Is the amount and quality of supplementary material appropriate?  N/A.

Specific comments, suggestions and questions

In general, all acronyms must be spelled out in their first appearances. Acronyms such as C-APC, C-FMR, C-CD (P.2, l.31) must be spelled out in the main text, since the abstract is not regarded as a part of the main text.

The acronyms are spelled out in the introduction of the revised manuscript (lines 33-36).

Almost all subscripts to mathematical symbols are italicized in the text. Such identifiers should not be italicized unless they are variables. (Example: Subscript 'noatt' to 'sigma' should not be italicized.) Similarly, the SI symbol for second is roman letter 's'. It should not be italicized. Many s in ms-1 are italicized in the text (e.g., page 15, lines 339, 341, 346 and 347).

The revised manuscript was modified according to this suggestion.

P.1, L 16, "demonstrate" <- 'demonstrates"

The revised manuscript was modified as suggested (line 16).

P.2, l 22, What are "the three instruments'?

This sentence was rewritten in the revised manuscript (lines 23-24).

P.2, l 26, Add "than CloudSat" at the end of the sentence. (line 27).

P.2, l 27, Add "the minimum detectable radar reflectivity factor is" before "-36 dBZ". (lines 28-29).

P.2, l. 29, "L2B" -> "L2b" (line 30).

P.2, l.32, "is" -> "are" (line 35).

P.2, l.33, "C-ATC"-> "C-APC" (line 36).

P.2, l.37, "(van Zadelhoff et al., 2022)" -> "van Zadelhoff et al. (2022)." (line 40).

P.2, l.39, "CPR On board processing" <- 'CPR onboard processing" (line 42).

The revised manuscript was modified as suggested.

P.2, l.52, "The return signal from each pulse results to another pair of I/Q at each range gate that includes contributions from the atmosphere (signal) and the radar receiver (noise)." The meaning of this sentence is obscure. What do the authors mean by "another pair of I/Q"?

This sentence was reworded to clarify its meaning (lines 58-59).

P.3, l.1, What are "the CPR Doppler radar moments"? Aren't they the Doppler spectral moments? This sentence was reworded for clarification (lines 61-62).

P.3, l.59, "21-22 consecutive I/Q pairs". According to the description a few lines below this expression, 22 pulses are used to estimate R(r,tao). Only 21 I/Q pairs can be made from 22 pulses. How does the processing unit use 22 consecutive I/Q pairs?

This was corrected in the revised manuscript (lines 65-68).

P.3, l.70, equation (3), R(r,tau) -> |R(r,tau)| , needs to take the absolute value.

The revised manuscript was modified according to this suggestion (line 79).

P.3, l.73, "angle" -> "angles"

P.3, l.73, "the velocity of the satellite along the flight direction, in the direction orthogonal to the orbit plane and the nadir direction." What does this phrase mean? Does it mean "the velocities of the satellite along the flight direction, in the direction orthogonal to the orbit plane and the nadir direction." (Shouldn't the "velocity" be plural?) If so, isn't the velocity in the direction orthogonal to the orbit plane always zero by definition? Or do the authors refer to the motion of the orbit plane relative to the rotating Earth?

This sentence was reworded as following (lines 80-83) to address these comments:

*"...the JAXA L1b CPR data product (called C-NOM) will include detailed geo-location information including the pitch, roll and yaw angle of the satellite, and the satellite velocity components along the flight direction, the direction orthogonal to the orbit plane and the nadir direction."*

P.3, l.79, "as a function of along-track distance, the hydrometeor-induced path integrated attenuation (PIA)." -> "as a function of along-track distance, and the hydrometeor-induced path integrated attenuation (PIA)."

The revised manuscript was modified according to this suggestion (lines 88-89).

P.4, Figure 1 (c), Why do the Z values at distance between 3500 and 3600 and height below 2 km larger than the corresponding Z in (b)?

Because the radar reflectivities reported in the top panel are attenuated reflectivities (by hydrometeors and gases) which the C-FM output has radar reflectivities corrected for gaseous attenuation.

P.5, l.94, "This explains the missing hydrometeor locations in the low levels around 3780 - 3800 and 4050 - 4180 km" The intervals specified by this sentence must be wrong. They must be around 3740-3760 and 4070-4130.

This sentence was reworded as follows in the revised manuscript (lines 103-105):

*"In some cases, the hydrometeor-induced attenuation can result to a complete extinction of the radar signal and loss of information. This is clearly visible by the lack of hydrometeor echoes in the low levels around 3740 - 3760 and 4070 - 4130 km."*

P.5, l.100, "detection's ." -> "detections." (remove the apostrophe and the extra space after 's'.) (line 110).

P.5, l. 104, "detection's ." -> "detections." (remove the apostrophe.) (line 114).

P.5, l. 108, "Path Integrating Attenuation" -> "Path Integrated Attenuation" (line 118).

The revised manuscript was modified as suggested.

P.5, l. 117, "measured surface echo" -> "measured surface cross section".

The sentence was reworded in the revised manuscript (line 128).

P.6, l.122 and l. 123, What does X-MET stand for? Which is the correct form, "X-MET" or "X-Met"?

The X-MET definition was added in lines 132-133 of the revised manuscript.

P.6, l. 124, "sigma_0" -> "sigma_{clr}"(line 135).

P.6, l.128, "range resolution" -> "range sampling interval" (line 139).

The revised manuscript was modified as suggested.

P.6, In figure captions to Fig. 2, Fig. 3, Fig. 4, and l. 157, "C-PRO" is not defined.

Figure captions and l. 157 were rewritten in the revised manuscript.

P.7, Fig. 3, The figure caption needs a period at the end of the caption.

A period was added at the end of this figure caption.

P.7, Fig. 3, Some explanation is required for the plot of the PIA and True PIA at the top of the figure. It seems to be a histogram.

The revised manuscript was modified to include this (lines 143-144).

P.8, l. 169, "… Battaglia et al., 2013; ?)" Replace the question mark with a meaningful word.

This was corrected in the revised manuscript (line 189).

P.8, l. 172, "between 6100 and 7400 Hz". This interval of PRF does not agree with the interval "between 6.2 and 7.4 kHz" written on page 2, line 51.

This was corrected in the revised manuscript (line 54).

P.9, Fig. 4, The figure caption needs a period at the end of the caption.

A period was added at the end of this figure caption.

P.9, Fig. 4, What are the conditions for 'Strong MS' and 'MS'? No regions marked by MS appear in Fig. 4.

In the case of strong MS conditions, the maximum in the radar reflectivity within +/-1000 m from the Earth's surface is not at the surface range gate. In the case of MS conditions, the maximum in the radar reflectivity within +/- 1000 m is at the surface range gate.

P.10, l. 185, "The JAXA CPR L1b data product (C-NOM), includes satellite …" -> "The JAXA CPR L1b data product (C-NOM) includes satellite …" Delete the comma after (C-NOM). (line 205).

P.10, l. 203, "Tanelli et al. (2002); Kollias et al. (2022) have …" -> "Tanelli et al. (2002) and Kollias et al. (2022) have …" (line 223).

P.10, l.206, "Others configurations" -> "Other configurations" (line 226).

P.10, l. 208, "Each point x0 in the along track direction" -> "Each point x0 in the along-track direction from the beam center"(line 228).

P.10, l.209, "velocity vSAT" -> "along-track velocity vSAT" (The altitude of the satellite may change with time. This vertical velocity is not included in equation (7) (line 228).

P.11, l.218, "Tanelli et al. (2002); Sy et al. (2014)" -> "Tanelli et al. (2002) and Sy et al. (2014)" (line 238).

P.11, l.228, "the complex R(tau) lag-one of the radar complex signal" -> "the lag-one autocovariance R(tau) of the radar complex signal" (line 247).

P.11, l.242, "exceeds VN." -> "exceeds the phase that corresponds to VN."(line 262).

P.12, l.258, "eta=1", In order to be consistent with this statement, the negative sign before eta in equation (13) should be deleted and "eta is an integer (0, 1, 2, …)" in line 244 should be modified to "eta is an integer (…, -2, -1, 0, 1, 2, …)". (line 262 and equation 13).

P.12, l. 262, "Signal-to-Noise" -> "signal-to-noise"(line 286).

The revised manuscript was modified as suggested.

P.12, l.268, What are the C-CLD and ACM-CAP algorithms?

These acronyms are spelled out in the revised manuscript (lines 292-293).

P.12, l.274, "Lx (km)", "Lz (km)" -> "Lx", "Lz". Delete unnecessary (km) to be consistent with the statement in line 278 in which Lz is specified in m

 The revised manuscript was modified as suggested (line 299).

P.13, l.289, "Fig.5a shows an example of the reflectivity-weighted hydrometeor sedimentation Doppler velocity." The sedimentation velocity is the sum of the hydrometer fall velocity and the vertical air velocity. There seems to be no negative sedimentation velocity regions in Fig. 5a.

The sedimentation velocity is not the Doppler velocity, is only the hydrometeors fall velocity. As such, it is always positive with positive Doppler velocities indicating motion towards the Earth's surface.

Doesn't the GEM model include any significant updraft area? (This question is also applicable to P.15, l.331, "The SVBE estimates are always positive.")

By definition the SVBE are always positive since they represent the best estimate for the sedimentation velocity of hydrometeors. In areas with significant updrafts, we can't retrieve the SVBE.

P.13, l.300, "affects" -> "affect" (line 325).

P.15, l.326, "the probability of finding 5 CPR Values" -> "the probability of finding at least 5 CPR Values" (lines 350-351).

P.15, l.335, "Each estimate is compare against" -> "Each estimate is compared against" (line 359).

The revised manuscript was modified as suggested

P.15, l.341, "2.5 ms−1 Fig. 6a." -> "2.5 ms−1 (Fig. 6a)."  "s" in ms-1 should not be italic. (This comment is applicable to several other places.).

This was corrected throughout the revised manuscript.

P.15, l.346, "1.8 ms-1". Judging from the data in Fig. 6, this velocity should be 1.6 ms-1. (line 370).

P.17, l.369, "(Battaglia and Kollias, 2014b)" -> "Battaglia and Kollias (2014b)" (line 393).

The revised manuscript was modified as suggested

P.17, l.372, " ice clouds radar reflectivity weighted mean Doppler velocity". Awkward.

This sentence was reworded to clarify its meaning (lines 395-396).

P.17, l.374, "flow chart" -> "flowchart" (line 398).

P.17, l.374, "The C-APC required input data are: the JAXA CPR L1b C-NOM and the X-MET files." -> "The input data to C-APC are the JAXA CPR L1b C-NOM and the X-MET files." (lines 398-399).

P.17, l.381, "Kalesse and Kollias (2013))" -> "Kalesse and Kollias (2013)" (lines 404-405).

The revised manuscript was modified as suggested.

P.19, l. 394-399, The reviewer was not able to understand this paragraph well enough. What does "each 1/8 of an orbit long data files" mean? Is it 'each data file that includes 1/8 of an orbit?

Yes, the EarthCARE data are packaged in frames and each frame is equivalent to 1/8 of an orbit.

P.19, l.412, "more noisy" -> "noisier"

The revised manuscript was modified as suggested (line 436).

P.19, l.415, "polynomial fit". Why is the harmonic fitting used in the case of Earth's surface, but the polynomial fitting used in the case of ice clouds?

We would like to apologize to the reviewer for the unfortunate wording. A harmonic fit is applied in the case of the Earth's surface and for ice clouds. After the harmonic fit, a polynomial fit is applied to remove additional residual biases that are not captures by the low order harmonic fit. In the revised manuscript, the parenthesis content is removed. At this point of the CPR algorithm development, we do not have any high frequency residual biases to remove. The

ability to perform a polynomial fit has been added as an extra processing step in case we diagnose during the commissioning phase that our harmonic model is not able to capture the observed CPR antenna mispointing characterization.

P.21, l.435 and 441, "L2A" -> "L2a" (lines 459 and 466).

P.21, l.435, "The CPR feature mask and reflectivity (C-FMR) product physical basis and algorithm structure is …", Do the authors mean by this subject "The physical basis and algorithm structure of the C-FMR product are …"? (line 459-460).

P.21, l.437, "and in combination with the improved sensitivity is expected to lead to more detections of low-level oceanic clouds (Burns et al., 2016)." What is the subject of this sentence? (line 462)

P.21, l.444, "however," -> "However," (line 469).

P.21, l.449, "the SVBE, that" -> "the SVBE that" (line 474).

P.21, l.459, "will be" <- 'is" (line 484).

The revised manuscript was modified as suggested.

---

## Author Comment (AC2)

RC1: 'Comment on egusphere-2022-1284', Referee #2

Reviewed by Matthew Lebsock

This paper describes three radar-only algorithms for the upcoming EarthCARE mission. The algorithms presented include a vertical feature mask, a Doppler correction product, and a pointing characterization product. The presentation is technically correct and relatively straightforward. The paper will provide an important citation for the at-launch product suite. I have only minor revisions regarding a few details of the presentation and some missing citations.

The authors would like to thank Matthew Lebsock for his useful and insightful feedback. A point-by-point response to the reviewer's comments is provided below.

Line 25: add 'Cloud Profiling Radar' or 'radar' after CloudSat. (line 26).

Line 25: change 'compare' <- 'compared' (line 26).

The revised manuscript was modified as suggested.

Section 2: It would be useful for many readers to understand the relationship between the three algorithms described here within the larger suite of EarthCARE products. Can an algorithm flow chart be incorporated? Or at least in words described?

The manuscript is part of an AMT special issue on the EarthCARE mission. There is another paper contribution (Wehr et al., 2022) that describe the mission and the EarthCARE L2 data production model. In the preparation of the manuscript, we were given specific instructions to avoid any repetition and refer to other manuscripts in the special issue that contain any needed material. Once we are close to the finalization of the special issue, we will make sure that the proper references are included in the manuscript to provide the necessary background.

Lines 93-95: The non-expert reader is not going to know what you are referring to. Add wording to the effect of 'beneath the convection near 4100 km'.

In the revised manuscript the following sentence was added: "In some cases, the hydrometeor-induced attenuation can result to a complete extinction of the radar signal and loss of

information. This is clearly visible by the lack of hydrometeor echoes in the low levels around 3740 - 3760 and 4070 - 4130 km."

Line 112-114: Mention the small temperature dependance.

The revised version of the manuscript was modified according to this suggestion (lines 122-123).

Line 114: Cite Lebsock et al., 2011 https://doi.org/10.1175/2010JAMC2494.1,

Citation added in line 124.

Lines 122 and 123: Is it X-MET or X-Met?

Correction made in line 133 of the revised manuscript.

Lines 140-145: Lebsock and Suzuki, 2016 (https://doi.org/10.1175/JTECH-D-16-0023.1) discuss the errors in this approach including (1) attenuation by undetected clouds, (2) systematic differences between water vapor in clear and cloudy columns, and (3) non-uniform beam filling (NUBF). The first two are small for the shallow subtropical cumulus clouds where this approach is best implemented. NUBF errors can be significant.

Section 3.2 Regarding Non-uniform beam filling errors for PIA - Even if you estimate a perfect PIA (averaged over a footprint) you have to translate that PIA into a TWP. The NUBF changes the relationship between PIA and TWP which can introduce significant errors. I understand the product won't produce a TWP but this limitation in the utility of PIA for deriving TWP deserves mention somewhere in the PIA section.

Section 3.2 You should mention somewhere in this section that MS signals frequent in stronger precipitation are often going to bias the PIA estimate low.

The revised manuscript was modified as follows to include this important information (lines 148-154):

*"In addition to the uncertainty introduced in the LWP estimation by the PIA measurement uncertainty, Lebsock and Suzuki (2016) discussed additional error sources including 1) attenuation by undetected clouds, (2) systematic differences between water vapor in clear and cloudy columns, and (3) non-uniform beam filling (NUBF). The first two are small for the shallow subtropical cumulus clouds where this approach is best implemented. On the other hand, the NUBF errors can be significant. Battaglia et al., 2020b discussed in detail the significant errors*

*that can be introduced in the LWP estimation by NUBF conditions. Another source of uncertainty is the presence of multiple scattering (section 3.3) that can cause biases in the PIA estimation."*

Line 154: Cite MS model.

The MS model citation was revised in the revised version of the manuscript (lines 171-172 ).

Line 169: '?' as a reference.

Correction made in line 189 of the revised manuscript.

Section 4.1.2: This section is too general. You don't describe the specific EarthCARE algorithm. Can you provide some details here on how you do the unfolding?

The last paragraph of this section was re-written and the formula used is now clearly stated in the revised version of the manuscript (lines 276-282).

Line 341 add '()' around 'Fig 6a' (line 365).

Line 432: add 'than cloudsat cpr' (line 456).

 The revised manuscript was modified as suggested.

Citation: https://doi.org/10.5194/egusphere-2022-1284-RC2

---

## Author Comment (AC3)

RC1: 'Comment on egusphere-2022-1284', Anonymous Referee #3

Review of: Processing reflectivity and Doppler velocity from EarthCARE's cloud profiling radar: the C- FMR, C-CD and C-APC products, Pavlos Kollias et al.

Overall Recommendation: Publish with minor changes

The authors would like to thank the reviewer for their useful and insightful feedback. A point-by-point response to the reviewer's comments is provided below.

Specific Comments:

- Line 21: Is it a good idea to mention a launch date within the paper?

We agree with the reviewer that typically (and particularly in the case of the EarthCARE mission that has been delayed for more than 10 years) is not a good practice to mention a launch date. However, at this point, there is high level of certainty within ESA that the EarthCARE launch will take place in 2024.

- Line 52: Maybe mention already why this frequency varies and in which way (intentionally, technically,...).

The revised manuscript was modified as follows to address this (lines 55-58):

*"Low PRF is used in the tropics and subtropics where the troposphere is deeper (18-20 km) and we need to space far apart in time the CPR pulses to avoid second trip echoes. At higher latitudes, the troposphere is shallower (10-12 km) and a higher PRF is possible. The PRF setting is very important since it determines the number of samples available for integration and affect the quality of the Doppler velocity measurements (Kollias et al., 2014)."*

- Line 70: A lot of a-priori knowledge is assumed. Estimation from autocovariance could actually be considered mathematical trivial, but a reference to a paper highlighting the technique would be really fruitful at this position. Actually, this technique only works if spectral leakage is avoided (i.e. the spectrum is fully and unambiguously recorded). So it should at least be shortly mentioned that it actually is applicable in the presence of all the adverse effects on the velocity measurement (NUBF etc...) and that it is representing the actual spectral width of the final velocity spectrum.

The revised manuscript was modified as follows to address this (lines 74-76):

"*The lag 0 and lag 1 autocovariance estimates are used for the estimation of the CPR Doppler moments using the pulse-pair moment estimator technique (Doviak and Zrnic, 1993).*"

Doviak, R. J. and Zrni´c, D. S.: Doppler Radar and Weather Observations, Academic Press, 1993

- Line 164: dBZint: please specify if this represents integrated Z values or the integrated linear values. I'm questioning if it is really necessary to define a new unit (dbZint) if a representation in actual physical units would be available.

We remove any reference to the dBZint parameter in the revised manuscript.

- Figure 4: The classification of "Strong MS" is retrieved for areas where there is no signal in the ideal simulation (at around 2700km below 5km). It should be noted in the figure that it depicts the multiple scattering influence on the measured values and not the multiple scattering originating at the given location. This disambiguation is implicitly made in the text, but it should also be contained in the figure.

The revised manuscript was modified as follows to address this (lines 183-185):

"*While the MS occurred above the height where I(z) exceeds 41 dB, its impact on the CPR observables is negligible above that height. The MS flag shown in Fig. 4c indicates the CPR ranges where the MS has a significant effect on the CPR observables.*"